# Design, Synthesis and Various Bioactivity of Acylhydrazone-Containing Matrine Analogues

**DOI:** 10.3390/molecules28104163

**Published:** 2023-05-18

**Authors:** Wanjun Ni, Hongjian Song, Lizhong Wang, Yuxiu Liu, Qingmin Wang

**Affiliations:** State Key Laboratory of Elemento-Organic Chemistry, Research Institute of Elemento-Organic Chemistry, College of Chemistry, Frontiers Science Center for New Organic Matter, Nankai University, Tianjin 300071, China; 2120140672@mail.nankai.edu.cn (W.N.); songhongjian@nankai.edu.cn (H.S.); wlzhong@nankai.edu.cn (L.W.)

**Keywords:** matrine, acylhydrazone, anti-TMV activity, insecticidal activity, fungicidal activity

## Abstract

Compounds with acylhydrazone fragments contain amide and imine groups that can act as electron donors and acceptors, so they are easier to bind to biological targets and thus generally exhibit significant biological activity. In this work, acylhydrazone fragments were introduced to the C-14 or C-11 position of matrine, a natural alkaloid, aiming to enhance their biological activities. The result of this bioassay showed that many synthesized compounds exhibited excellent anti-virus activity against the tobacco mosaic virus (TMV). Seventeen out of 25 14-acylhydrazone matrine derivatives and 17 out of 20 11-butanehydrazone matrine derivatives had a higher inhibitory activity against TMV than the commercial antiviral agent Ribavirin (the in vitro activity, in vivo inactivation, curative and protection activities at 500 µg/mL were 40.9, 36.5 ± 0.9, 38.0 ± 1.6 and 35.1 ± 2.2%, respectively), and four 11-butanehydrazone matrine derivatives even had similar to or higher activity than the most efficient antiviral agent Ningnanmycin (55.4, 57.8 ± 1.4, 55.3 ± 0.5 and 60.3 ± 1.2% at 500 µg/mL for the above four test modes). Among them, the *N*-benzyl-11-butanehydrazone of matrine formed with 4-bromoindole-3-carboxaldehyde exhibited the best anti-TMV activity (65.8, 71.8 ± 2.8, 66.8 ± 1.3 and 69.5 ± 3.1% at 500 µg/mL; 29, 33.5 ± 0.7, 24.1 ± 0.2 and 30.3 ± 0.6% at 100 µg/mL for the above four test modes), deserving further investigation as an antiviral agent. Other than these, the two series of acylhydrazone-containing matrine derivatives were evaluated for their insecticidal and fungicidal activities. Several compounds were found to have good insecticidal activities against diamondback moth (*Plutella xylostella*) and mosquito larvae (*Culex pipiens pallens*), showing broad biological activities.

## 1. Introduction

Acylhydrazone compounds contain both amide and imine groups. Therefore, they can serve as hydrogen bond donors or acceptors. They have a strong ability to coordinate, have multiple coordination modes and it is easy to adjust their orientation in space, so they easily form stable complexes with biological targets, thus affecting their biological activities. From this point of view, acylhydrazone is considered a privileged group in the structural modification of new drugs [1,2]. A variety of acylhydrazone-containing compounds have been reported to have pharmaceutical activity [3,4,5,6,7,8,9,10]. In recent years, there has been an increasing number of reports on the activities of such compounds against agricultural diseases and pests. For example, β-tetrahydrocarboline acylhydrazone derivatives have excellent antiviral activity against the tobacco mosaic virus (TMV), good fungicidal activity and the most potent anti-TMV compound among them (later named chloroinconazide [11]) exhibited excellent anti-TMV activity in the field [12]. Moreover, oxindole spirocyclic acylhydrazone derivatives [13], trans-ferulic acid [14], indole [15], carbazole [16], tryptophan [17] and azepino [4,5-b]indole acylhydrazone derivatives [18] were also identified as novel anti-TMV agents. In addition, C8-acylhydrazone coumarin-type derivatives of osthole exhibited better insecticidal activity against Mythimna separata Walker [19], and quinoline derivatives with acylhydrazide were found to have novel antifungal activity [20] (Figure 1).

Matrine is widely present in the leguminous plants *Sophora flavescens*, *Sophora alopecuroides* and *Sophora subprostrara*. Matrine and its derivatives have been reported to have various biological activities, such as anticancer, anti-inflammatory, antioxidant, insecticidal and bactericidal activities [21,22,23,24,25,26,27]. It is worth mentioning that matrine was found for the first time to have anti-TMV activity by our group [28,29].

There are mainly two types of artificial derivations of matrine. One is to preserve the tetracyclic parent structure of matrine and introduce an alkoxy or amino group, a double bond, acyl group, amino group, oxime, etc. at the C-13 or C-14 position, convert the amide carbonyl at the C-15 position into an imine, or form a heterocycle on its D ring [30,31,32,33,34,35,36]; the other is to open its D ring and then introduce carboxyl, ester, amide, etc. on the C-11 side chain and an alkyl, acyl, sulfonyl, etc. on the exposed nitrogen [30,37,38,39,40,41,42,43,44]. In this paper, in order to improve the physicochemical properties and enhance the biological activity of matrine, we introduced a unique acylhydrazone fragment into the C-14 position of matrine or the C-11 position of ring-opened matrine (Figure 1) and the two series of matrine derivatives were studied for their anti-TMV, fungicidal and insecticidal activities.

## 2. Results and Discussion

### 2.1. Chemistry

#### 2.1.1. The Synthesis and Configuration of 14-Acylhydrazone Derivatives of Matrine

As depicted in Figure 1, matrine was first treated with lithium diisopropylamide (LDA) at −78 °C and then reacted with dimethyl carbonate to obtain 14-acylated matrine **A** with a pair of diastereomers due to the newly formed chiral carbon (C-14). The hydrazination step was carried out using 80% hydrazine hydrate under microwave irradiation and a pair of 14-acylhydrazine derivatives of matrine **B** were obtained in a quantitative yield. Compound **B** was then reacted with different aldehydes or ketones in refluxing toluene, providing the desired 14-acylhydrazone derivatives of matrine **1**–**25**. The products were purified by column chromatography with a yield range of 54% to 80%.

Theoretically, acylhydrazone compounds should contain multiple isomers. Actually, from the ^1^H NMR spectra of compound **20**, which was selected as an example (Figure 2), we could clearly observe four N-H peaks appearing around 11 ppm and four C-H peaks (7.1–7.6 ppm) belonging to imine. Similarly, compound **13** also showed as being a mixture of four isomers. According to the literature [45] and our previous research results [12,13], we believe that the four configurations were mainly induced by the chirality of C-14 and synperiplanar E/antiperiplanar E configuration of amide. Because the four isomers were similar in polarity, they could not be separated from each other by column chromatography.

#### 2.1.2. The Synthesis and Configuration of 11-Butanehydrazone Derivatives of Matrine

To open the D ring, matrine was first treated with a sodium hydroxide aqueous solution and then hydrochloric acid to provide an 11-butanic acid derivative of matrine **C** as a white solid, which could be obtained by filtration (Figure 2). Next, benzyl chloride and **C** were mixed in the presence of excess potassium carbonate, affording a di-benzylation product **D**. Compound **D** then underwent hydrazination with 80% hydrazine hydrate in refluxing ethanol to obtain the 11-butanehydrazine **E**, which was reacted with different aldehydes to finally afford a series of the corresponding 11-butanehydrazone analogues of matrine **26**–**45**.

These series of matrine derivatives generally gave rise to 4 N-H peaks around 11 ppm (**34** as an example, Figure 3), indicating that there were also multiple isomers. We came to the conclusion that this not only took into account the conformers of synperiplanar E and antiperiplanar E, but also considered the chiral factor on nitrogen. We also noticed that some of the derivatives had several additional small peaks at 9.5–10.5 ppm and a small peak at 4.8 ppm (**39** as an example), indicating the possibility of keto/enol tautomerism. These isomers also could not be separated from each other because of their similar polarity.

However, if we consider the keto/enol tautomerism of compounds **1**–**25**, which seem more prone to form enol isomers than compounds **26**–**45**, their isomers should have far exceeded four. We could not reasonably explain why we obtained only four of them. We designated the four isomers as caused by chiral C-14 and conformational isomerism (as shown in Figure 2), but we did not completely exclude them as chiral C-14 and ketone/enol tautomers.

### 2.2. Anti-TMV Activity

Tobacco mosaic virus (TMV) is a plant virus that is extremely harmful to agricultural production. Matrine and its derivatives were found to have good anti-TMV activity in our previous work [28,29]. We expected that the introduction of a hydrazone fragment into matrine could result in better activity. The structures of the synthesized matrine derivatives **1**–**45** are provided in Figure 1 and Figure 2. We applied four test modes to study the antiviral activity, including in vitro mode (using tobacco leaves in vitro), in vivo (using potted tobacco plants) inactivation, curative and protection modes. The detailed methods are described in the Appendix A.

#### 2.2.1. The Anti-TMV Activity of 14-Acylhydrazone Derivatives of Matrine **1**–**25**

Generally, the anti-TMV activities in vitro of most 14-acylhydrazones **1–25** were higher than that of matrine (inhibition rate 29.8%, 500 μg/mL), and nearly half of them were equivalent to or higher than that of the commercialized virucide Ribavirin (40.9%, 500 μg/mL) (Table 1). One compound, 5-methyl-2-thiophenecarboxaldehyde matrine-14-carbonylhydrazone (**25**), showed the highest inhibitory activity in vitro (55.4%, 500 μg/mL), which was similar to that of commercialized virucide Ningnanmycin (55.4%, 500 μg/mL).

Most 14-acylhydrazones of matrine that had good anti-TMV activity in vitro also had good activity in vivo. Seventeen out of the 25 14-acylhydrazones exhibited a higher inhibitory activity than Ribavirin (the inhibitory activities in the inactivation, curative and protection modes were 36.5 ± 0.9, 38.0 ± 1.6 and 35.1 ± 2.2%, respectively, at a concentration of 500 µg/mL), but the structure–activity relationship was not so clear. Among the 14-acylhydrazones from mono-substituted benzaldehydes, 4-trifluoromethoxybenzaldehyde hydrazone (**14**) and 3-nitrobenzaldehyde hydrazone (**15**) exhibited higher inhibitory activity; among the 14-acylhydrazones from multi-substituted benzaldehydes, 3-nitro-4-hydroxy-benzaldehyde hydrazone (**9**) exhibited the highest inhibitory activity; the two heteroaryl aldehyde hydrazones **24** and **25** also had decent activity, but all of the compounds mentioned above were lower than Ningnanmycin. Alkyl aldehyde hydrazones (**20**–**23**) had relatively inferior activity. Branched alkyl aldehyde and cyclic alkyl aldehyde hydrazones (**21** and **23**) had better activities than linear aldehyde hydrazones (**20** and **22**), and the hydrazone **22**, synthesized from long chain alkyl aldehyde, had the lowest activity.

#### 2.2.2. The Anti-TMV Activity of 11-Butanehydrazone Derivatives of Matrine **26**–**45**

Most of the 11-butanehydrazone derivatives of matrine **26**–**45** had better anti-TMV activities than the 14-acylhydrazone derivatives **1**–**25** (Table 1). Seventeen out of the 20 11-butanehydrazone derivatives exhibited higher inhibitory activities than both matrine and Ribavirin, among them four compounds (**33**, **36**, **42** and **43**) had higher in vitro activity and six compounds (**33**, **34**, **36**, **40**, **42** and **43**) had higher in vivo activities than Ningnanmycin at the concentration of 500 μg/mL (the inhibitory activities of Ningnanmycin on the in vitro mode activity, in vivo inactivation, curative and protection modes were 55.4, 57.8 ± 1.4, 55.3 ± 0.5 and 60.3 ± 1.2%, respectively).

For the mono-substituted benzaldehyde hydrazone derivatives, compounds with electron-withdrawing groups on the benzene ring were higher than those with electron-donating groups on the benzene ring no matter what the group was on the ortho or meta position. For example, 2-trifluoromethylbenzaldehyde hydrazone (**36**) had much better in vitro and in vivo activities than 2-hydroxybenzaldehyde hydrazone (**29**). Similarly, the meta-bromo, -cyano and -nitro substituted benzaldehyde hydrazones (**34**, **35**, and **37**, respectively) had higher activities than meta-hydroxybenzaldehyde hydrazone (**30**). For di-substituted or multi-substituted benzaldehyde hydrazones, the presence of a meta substituent on the benzene ring always helped improve the inhibitory activities. For example, the order of the activities was 3,4-dimethoxy benzaldehyde derivative (**27**) > 2,4-dimethoxy derivative (**26**), 2-hydroxy-5-bromo derivative (**31**) > 2-hydroxy derivative (**29**) and 3,4-dichloro derivative (**33**) > 4-dichloro derivative (**32**).

For the hydrazones formed with naphthyl aldehyde and heteroaryl aldehyde, the activities varied widely. 3-Pyridyl aldehyde hydrazone (**41**) had the least activity, thiophene-3-carboxaldehyde hydrazone (**40**) exhibited similar activity to Ningnanmycin, while 4-bromoindole-3-carboxaldehyde hydrazone (**42**) (65.8, 71.8 ± 2.8, 66.8 ± 1.3 and 69.5 ± 3.1%, 500 μg/mL; 29, 33.5 ± 0.7, 24.1 ± 0.2 and 30.3 ± 0.6%, 100 µg/mL) displayed the best anti-TMV activity among all the synthesized compounds both in vitro and in vivo. Therefore, the anti-TMV activity of hydrazones synthesized from substituted thiophene-3-carboxaldehydes and substituted indole-3-carboxaldehydes deserves further study.

In the class of alkyl aldehyde hydrazone derivatives, the activities of the hydrazone derived from straight chain alkyl aldehyde was better than those derived from branched aldehydes. As indicated in Table 1, *n*-octylaldehyde hydrazone (**43**) had better activities than cyclohexyl formaldehyde hydrazone (**45**) and trimethylacetaldehyde hydrazone (**44**).

### 2.3. Insecticidal/Acaricidal Activity

Matrine and many acylhydrazones have been reported to have a variety of biological activities, so we screened the synthesized matrine hydrazone derivatives **1**–**45** for insecticidal and acaricidal activities, including larvicidal activities against the oriental armyworm (*Mythimna separate*), cotton bollworm (*Helicoverpa armigera*), corn borer (*Ostrinia nubilalis*), diamondback moth (*Plutella xylostella*) and mosquito larvae (*Culex pipiens pallens*), aphicidal activity against the aphid (*Aphis medicnginis* Koch) and acaricidal activity against the spider mite (*Tetranychus cinnabarinus*). The bioassay methods and partial data are described in the Appendix A.

From the data we found that almost all matrine hydrazone derivatives had no fatal activities against the oriental armyworm, cotton bollworm, corn borer, aphid and spider mite (Appendix A). On the contrary, they had relatively good larvicidal activity against diamondback moth and mosquito larvae (Table 2). When the concentration was 100 µg/mL, there were seven 14-acylhydrazone derivatives (**9**, **11**, **12**, **14**, **18**, **20** and **23**) whose insecticidal activity was above 50%.

For the insecticidal activity against mosquito larvae, more than half of the 14-acylhydrazone derivatives exhibited a 100% mortality rate at a concentration of 10 µg/mL. Moreover, the mortality rate of 3-nitro-4-hydroxybenzaldehyde hydrazone (**9**) and 3-nitrobenzaldehyde hydrazone (**15**) still reached 40% at a concentration of 2 μg/mL.

In general, these compounds showed insecticidal activity against diamondback moth and mosquito larvae, indicating that the acylhydrazone derivatives of matrine, especially 14-acylhydrazones, have a wide range of biological activities.

### 2.4. Fungicidal Activity

The matrine hydrazone derivatives **1**–**45** were also evaluated for their fungicidal activity against 14 phytopathogens in vitro. From Table 3 and Appendix A we can see most derivatives had low fungicidal activity. In contrast, these compounds had relatively high activity against *Rhizoctonia solani*, with seven compounds (**3**, **4**, **10**, **15**, **16**, **21** and **40**) showing greater than 70% inhibitory activity at 50 µg/mL, but the activities were still lower than those of commercialized carbendazim and chlorothalonil.

## 3. Materials and Methods

### 3.1. Instruments and Chemicals

The reagents were purchased from commercial sources and were used as received. All anhydrous solvents were dried and purified by standard techniques just before use. Matrine was purchased from Baoji Biological Development Co., Ltd. (Xi’an, China). Compound **A** [28], **C** [46] and **D** [46] were synthesized as referred by the literature.

The reaction progress was monitored by thin-layer chromatography on silica gel GF254 with detection by UV. Microwave reaction was conducted in a microwave synthesizer (CEM Discover SP). Melting points were determined using an X-4 binocular microscope melting point apparatus and the thermometer was uncorrected. The ^1^H NMR spectra were obtained by using a Bruker AV 400 with CDCl_3_ or DMSO-d_6_ as a solvent. Chemical shifts (δ) were given in parts per million (ppm) and were measured downfield from internal tetramethylsilane. The ^13^C NMR spectra were recorded by using a Bruker AV 400 (100 MHz) with CDCl_3_ or DMSO-d_6_ as a solvent. Chemical shifts (δ) were reported in parts per million using the solvent peak as the standard. High-resolution mass spectra were obtained with an FT-ICR MS spectrometer (Ionspec, 7.0 T).

### 3.2. Synthetic Procedures

The synthetic routes are given in Figure 1 and Figure 2.

#### 3.2.1. Synthesis of (41S,7aS,13aR,13bR)-10-Oxododecahydro-1H,5H,8H-dipyrido[2,1-f:3′,2′,1′-ij][1,6]naphthyridine-11-carbohydrazide (**B**)

A mixture of **A** (3.0 g, 9.8 mmol), 80% hydrazine hydrate (3.0 mL, 49.0 mmol) and dry ethanol (8 mL) were added to a 35 mL microwave tube, and the reaction was conducted under microwave conditions of 80 °C at 100 W for 0.5 h. The solvent and excess hydrazine hydrate were removed under reduced pressure to obtain a pair of diastereomers **B** (3.2 g, yield 100%, dr 1: 1) as a yellow oil. ^1^H NMR (400 MHz, CDCl_3_) δ: 4.39–4.01 (m, 1H), 3.92–3.79 (m, 1H), 3.59 (brs, 2H, NH_2_), 3.28 (s, 0.5H, COCHCO), 3.16–3.02 (m, 2H), 2.81 (dd, J = 11.2, 10.8 Hz, 2H), 2.42–2.37 (m, 0.5H, COCHCO), 2.11–1.86 (m, 6H) and 1.73–1.35 (m, 10H). ^13^C NMR (100 MHz, DMSO-d_6_) δ: 169.5, 169.4, 165.9, 165.4, 63.1, 62.8, 56.6, 56.5, 52.7, 52.5, 47.5, 46.8, 42.8, 41.9, 41.6, 40.9, 35.3, 34.9, 27.4, 27.4, 25.9, 25.8, 25.2, 25.1, 22.5, 21.9, 20.7, 20.6, 20.2, 14.0 and 13.6. The HRMS (ESI) calculated for C_16_H_27_N_4_O_2_ [M + H]^+^ 307.2129 was found to be 307.2133.

#### 3.2.2. Synthesis of (4^1^S,7aS,13aR,13bR)-N′-((E)-2,4-Dimethoxybenzylidene)-10-oxododecahydro-1H,5H,8H-dipyrido[2,1-f:3′,2′,1′-ij][1,6]naphthyridine-11-carbohydrazide (**1**)

To matrine-14-carbohydrazide **B** (0.5 g, 1.6 mmol) in 50 mL toluene, 2,4- dimethoxy benzaldehyde (0.53 g, 3.2 mmol) was added. After the mixture was refluxed for 7 h, the solvent was removed under reduced pressure, and the residue was purified by column chromatography (V_DCM_:V_MeOH_ = 10:1) to obtain compound **1** (0.43 g, yield 60%, dr 1:1:1:1) as a light yellow solid. The Mp was 166−168 °C. ^1^H NMR (400 MHz, DMSO-d_6_) δ: 11.36 (s, 0.20H, NH), 11.35 (s, 0.20H, NH), 11.24 (s, 0.20H, NH), 11.15 (s, 0.20H, NH), 8.45–8.19 (4s, CH=N), 7.79–7.55 (m, 1H), 6.66–6.53 (m, 2H), 4.16 (s, 1H), 3.85–3.81 (m, 7H), 3.09–2.75 (m, 4H) and 2.33–1.29 (m, 17H). The HRMS (ESI) calculated for C_25_H_35_N_4_O_4_ [M + H]^+^ 455.2653 was found to be 455.2660.

Compounds **2**–**25** were synthesized according to the synthesis procedure of compound **1**.

#### 3.2.3. Synthesis of 4-((1R,3aS,3a1S,10aR)-2-Benzyldecahydro-1H,4H-pyrido[3,2,1-ij][1,6]naphthyridin-1-yl)butanehydrazide (**E**)

The mixture of N-benzyl-11-butanoyl benzyl ester of matrine derivative **D** (2.20 g, 5.90 mmol) and hydrazine hydrate (3 mL, 48.0 mmol) in 100 mL of dry ethanol was refluxed for 8 h until the reaction was complete, then the solvent was removed. Ethanol (5–10 mL) was added and evaporated three times to remove the excess hydrazine hydrate to obtain crude **E** (containing a small amount of benzyl alcohol) as an orange oil, which was used in the next step without further purification.

#### 3.2.4. Synthesis of 4-((1R,3aS,3a1S,10aR)-2-Benzyldecahydro-1H,4H-pyrido[3,2,1-ij][1,6]naphthyridin-1-yl)-N′-((E)-2,4-dimethoxybenzylidene)butanehydrazide (**26**)

The mixture of crude **E** (0.50 g, 1.40 mmol) and 2,4-dimethoxy benzaldehyde (0.46 mL, 2.80 mmol) in 30 mL of toluene was refluxed for 5 h. After the reaction was complete, the solvent was removed and then the residue was purified by column chromatography (V_DCM_:V_MeOH_ = 20:1) to obtain compound **26** (0.39 g, yield 66%, isomers mixture) as a yellow solid. The Mp was 139−141 °C. ^1^H NMR (400 MHz, DMSO-d_6_) δ: 11.45–11.03 (m, 1H, NH), 8.47–8.40 (m, 0.5H, CH=N), 8.24−8.19 (m, 0.5H, CH=N), 7.70–7.29 (m, 6H), 6.61–6.50 (m, 2H), 4.22–4.00 (m, 1H, PhC**H**_2_N), 3.83–3.79 (4s, 6H), 3.25–2.57 (m, 7H) and 2.33–1.23 (m, 18H). The HRMS (ESI) calculated for C_31_H_43_N_4_O_3_ [M + H]^+^ 519.3330 was found to be 519.3334.

Compounds **27**–**45** were synthesized according to the synthesis procedure of compound **26**. The physicochemical data and NMR spectra of compounds **1**–**45** are provided in the Appendix A.

### 3.3. Biological Assay

Detailed bioassay procedures for the anti-TMV, insecticidal and fungicidal activities were performed according to our published literature [28,29] and were also described in the Appendix A.

## 4. Conclusions

In summary, two series of matrine derivatives were obtained by introducing an acylhydrazone fragment into the C-14 or C-11 position of matrine and their pesticide activities were evaluated, including their anti-TMV, antifungal and insecticidal activities. In general, for the 14-acylhydrazones and 11-butanehydrazones, the type of aldehyde, the number, size and electronic property of the substituents on the benzene or heteroaryl ring all had a great influence on the activity of TMV. The 11-butanehydrazones exhibited relatively better anti-TMV activities. Four compounds (**33**, **36**, **42** and **43**) had equivalent or higher activities both in vitro and in vivo than Ningnanmycin at a concentration of 500 μg/mL, and compound **42** exhibited much higher activities than Ningnanmycin at both 500 μg/mL and 100 μg/mL, which deserves further investigation. In addition, several compounds were found to have good insecticidal activities against diamondback moth and mosquito larvae, showing broad biological activities. These studies will guide the further design and development of more potential bioactive derivatives of matrine hydrazone.

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
