# Peer review of "Design, Synthesis and Various Bioactivity of Acylhydrazone-Containing Matrine Analogues"

_molecules, 2023, doi:10.3390/molecules28104163_

Round 1
Reviewer 1 Report
Liu and Wang group reported syntheses of two types of acylhydrazone compounds derived from Matrine, a commercially available anti-cancer alkaloid and their bioassays for tobacco mosaic virus (TMV), and other insecticidal and fungicidal activities. The reviewer focuses on the synthesis section.
The stereochemical speculation based on 1H NMR measurement is one of the crucial synthetic issue of the manuscript. Achi group describes the study on E, Z-configurations of the related N-acylhydrazones [Ref. 45], in which rational assignment 1H and 13C NMRs were disclosed. Compared with the Achi group study, the presented results and discussions are somewhat ambiguous. The reviewer suggests that the description order in lines 81-91 should be exchanged, i.e. the first is that of compound 16 and the second is that of 20 for the readers’ easier understandings. Keto/enol tautomerization is the other important but complex issue. Can the methine proton (around ca. 3.5) and/or the enol proton (ca. 10.0-15.0) could be assigned?
Due to the absence of Keto/enol tautomerism, the case of compounds 34 and 36 is simpler. Accordingly, more concrete discussion may be possible.
On the whole, the reviewer recommends the publication in Molecules after major revisions commented above and below.
<Comments and suggestions>
1. Abstract, line 12: better → significant
2. Abstract, line 22: matrine (42) → intact matrine The numbering ## should not be addressed in the Abstract section due to its independency of the Abstract.
3. Figure 1: “⇨” means retro-synthesis. Chang the dotted arrow, for an example
4. P. 2, lines 57-63: Too long one sentence. Two “introducing” and two “introduce” appear in one sentence. More and more concise and clear description is required.
5. P. 3, line 74: … matrine A. A was … → … matrine A with …
6. P. 3, line 72 and Scheme 1: Instead of nBuLi / iPr2NH, LDA should be described, due to the simple and privileged protocol.
7. P. 3, line 85: four N-H peaks around 7.5 ppm → four N-H peaks (7.0-7.8 ppm)
8. P. 3, line 87: “as depicted in Figure 2” should be deleted due to the repetition.
9. Scheme 2, Numbers “16” and “20” should be embedded in the chart. Also in Scheme 3 (34, 36).
10. P. 4, line 99: Then carried out → underwent
11. P. 4, line 100: giving → to give
12. P. 4, line 101: corresponding → a series of the corresponding derivatives → analogues
13. P. 5, lines 104-106: the tense should be changed to passive-style
More concise and clear description will be required.
Reviewer 2 Report
The authors describe the synthesis and biological evaluation of a range of matrine derivatives bearing acylhydrazone moieties in the C-14 and C-11 positions. This work follows the previous publications of the authors on the same topic (J. Agric. Food Chem. 2017, 65(10), 2039-2047; Molecules 2022, 27(21), 7563).
The work is interesting and gives access to a great variety of novel matrine derivatives, some of which showing promising biological activity. Overall, the paper is well structured but still requires some improvements. Selected examples are given below.
- In section 2, results and discussion, some additional information regarding the general efficiency of the reactions must be given, namely in the synthesis of the target compounds 1-25.
- Schemes 1 and 2 must be corrected: the chemical formulas of the reagents must be replaced by the abbreviations (CH3CH2CH2Li to BuLi, etc), the reaction conditions (time, temperature) must be introduced, the R2 of all compounds 1-25 must be added as well as the yields of the isolated compounds.
- In section 2.2.2, some sentences must be improved for better understanding of the structure-activity results.
- In section 4.2 and SI, the described synthetic methodologies afforded 47 new compounds (intermediates B and E, compounds 1-45) which are not fully characterized regarding to their physical and chemical properties. Of the 47 compounds, only compound B presents the 13C NMR data and spectrum. The 13C NMR data and spectra of all new compounds must be included. These data are crucial to confirm the structural assignment of the isolated compounds. Also, some of the 1H NMR spectra presented in SI don’t seem to corroborate the structural assignment. For example, in the 1H NMR spectrum of intermediate E, the signal corresponding to the N-CH2Ph protons appears at 4.49 ppm as a singlet. The 1H NMR spectra of compounds 26-45, which contain the same N-benzyl group, should present the same signal without a significant variation of the chemical shift. However, that is not clear in some of the presented spectra (e.g. the 1H NMR spectra of compound 34, for example). Please confirm all the spectra presented in SI as well as the NMR assignment of all compounds.
- Also, the chemical shifts belonging to the NH and to the CH imine protons should be described differently. For example, for compound 1, instead of “11.36 (s, 0.20H), 11.35 (s, 0.20H)…” it should be “11.36 (s, NH), 11.35 (s, NH),…”.
- The nomenclature of all new compounds must be corrected according to the IUPAC guidelines.
- The conclusion section should also be improved.
Overall, the quality of the english language is fair. Nevertheless, some sentences should be improved as described above.
Round 2
Reviewer 1 Report
The revised manuscript has been reviewed now. Replacement of compound 13 to 16 is a better renewal due to its clear NMR chart. I would like to accept it after only one following revision.
As is commented in the first review, “Keto/enol tautomerization is the other important but complex issue. Can the methine proton (around ca. 3.5) and/or the enol proton (ca. 10.0-15.0) could be assigned?”
Please comment this issue.
Minor editing of English language will be required
